# Long-Term Chloride Accumulation on Concrete Surface in Marine Atmosphere Zone—Modelling the Influence of Exposure Time and Chloride Availability in Atmosphere

Gibson Rocha Meira [1,*] , Pablo Ramon Ferreira [1] and Carmen Andrade [2]

1 Paraíba Federal Institute, Civil and Environmental Engineering, Post-Graduate Program of Federal University of Paraíba (UFPB), Av. João da Mata, 256, João Pessoa 58015-020, Brazil; pablo.ferreira@ifpb.edu.br
2 International Centre for Numerical Methods in Engineering (CIMNE), Campus Nord UPC C/Gran Capità, S/N, 08034 Barcelona, Spain; candrade@cimne.upc.edu
* Correspondence: gibson.meira@ifpb.edu.br

**Abstract:** Surface chloride concentration ($Cs$) is a key parameter used to feed models adopted to simulate chloride penetration into concrete and evaluate the initial period of corrosion. Although there are several models that have been proposed for the representation of $Cs$ behaviour in the marine atmosphere zone, such models are still scarce. In this context, we analysed the behaviour of surface chloride concentration in concrete specimens exposed over 12.5 years in a marine atmosphere zone in the northeast of Brazil. The experimental work was carried out in two steps: environmental characterization, which was undertaken for temperature, relative humidity, rainfall, wind characteristics and sea-salt data; and chloride concentration measurements for the concrete surface considering three different concrete mixtures with w/b ratios of 0.65, 0.57 and 0.50. The results showed that the $Cs$ increase over time followed three stages: a first short stage characterised by an initial dispersion, followed by an increase period and then a final period of stabilisation, which was not fully reached in the present study. This behaviour can be represented by a power function or a sigmoidal function, with a better fit with the latter. Chloride concentration in the atmosphere plays an important role in $Cs$ behaviour. Higher availability of chlorides means higher $Cs$ values. The relationship between $Cs$ and the rate of chloride deposition on a wet candle was analysed and the function $Cs = C_0 + k_{cs} \cdot (Dac)^n$ was the one that best fit the experimental data.

**Keywords:** concrete; corrosion; marine atmosphere zone; surface chloride concentration





## 1. Introduction

Surface chloride concentration ($Cs$) is one of the main parameters used to feed models adopted to simulate chloride penetration into concrete structures [1,2]. As $Cs$ governs the availability of the chloride ions that are transported into concrete, representing its behaviour in a more accurate way should result in more accurate forecasting of the initiation period of corrosion (the time that elapses between the exposure of the reinforced concrete structure and the reinforcement depassivation).

Regarding $Cs$ behaviour over time, it has been observed that $Cs$ tends to increase as time progresses [3,4]. However, this increasing trend weakens over time and tends to become stabilised, which can be observed after about 10 years of exposure time in some cases [5]. Taking into account that literature data on $Cs$ behaviour over time are still scarce, Figure 1 presents available data from several authors focused on the marine atmosphere zone. Despite the short exposure time, it is possible to see signs of the weakening tendency previously mentioned for those cases where the exposure time was longer.

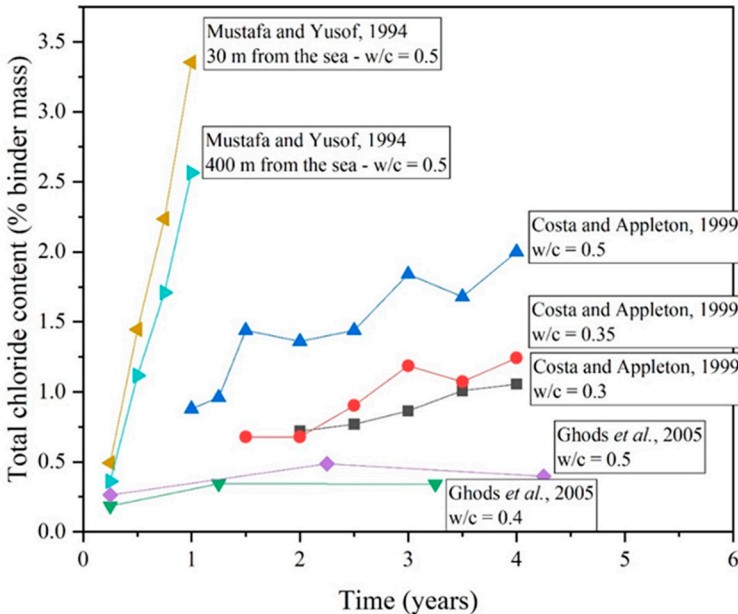

**Figure 1.** Relationship between surface chloride content in concrete and exposure time [3,6,7].

The literature thus shows that *Cs* behaviour can be represented by various mathematical models (power or exponential growth functions), which are presented in Table 1. All these functions proposed in the literature and shown in Table 1 are based on exposures in the marine atmosphere zone. From this table and from Figure 1, it can be observed that not many studies have focused on the behaviour of *Cs* in the marine atmosphere zone and that most proposed functions are power or exponential functions, with the first being predominant.

**Table 1.** Models from the literature used to represent the behaviour of surface chloride concentration in concrete in the marine atmosphere zone.

| Source of Data | Exposure Time (Years) | Function | Authors | Year |
|---|---|---|---|---|
| Japan | 23–58 | $Cs = at^{0.5}$ | Uji et al. [8] | 1990 |
| Portugal | 0.5–5.5 | $Cs = at^b$ | Costa and Appleton [3] | 1999 |
| United States of America | 2–16 | $Cs = a\,(a\,(1 - e^{bt}))$ | Kassir et al. [9] | 2002 |
| South Korea | 0.7–48.7 | $Cs = aLn(bt + 1) + c$ | Pack et al. [10] | 2010 |
| Data from the literature | 0–3 | $Cs = a + bt^{0.5}$ | Zhou et al. [11] | 2016 |
| Data from the literature | 0–5 | $Cs = a(1 - e^{bt})$ | Yang et al. [4] | 2017 |

$Cs$ = surface chloride concentration, $t$ = exposure time.

Following the analysis of the results presented by the studies referenced in Figure 1 and Table 1, it can be observed that, in general, *Cs* sharply increases in the initial years of exposure and, in the following years, this trend gives way to a more understated increase. Depending on the function adopted to represent behaviour of the data, the final part can assume a more asymptotic shape, which denotes a trend towards stabilisation over the years. Moreover, in some cases, *Cs* presents some fluctuation, which can be a result of environmental interaction [5,12]. This can be more easily seen in the initial years of exposure, when the chloride concentrations are still low and the impact of the environmental variables, such as rainfall, on chloride content on concrete surfaces can assume a stronger magnitude.

Regarding the shape of the *Cs* curve, continuous cement-paste hydration over time is one of the aspects that may influence behaviour. As hydration of cement paste advances,

the concrete surface becomes less porous and fewer chloride ions can be captured in this region [10,13]. As a consequence, the rate of the increase in *Cs* weakens. Furthermore, the concrete's ability to capture chlorides decreases with the increase in chloride concentration as a consequence of the increase in the bound chloride content, which also contributes, in the same way, to weakening the increase in *Cs* over time [12].

Another aspect to be considered in this analysis is the aggressiveness of the environment to which concrete structure is subjected. At locations with a greater availability of chlorides in the atmosphere, there is a stronger increase in *Cs* in the initial years, with a subsequent attenuation with time [3,14]. However, it can be expected that *Cs* may present some fluctuation over the years, which may be related to the movement of ions towards bulk concrete or to direct environmental interaction, which can promote several removal effects, such as surface chloride removal due to rainfall [12].

Although several models have been proposed to represent *Cs* behaviour over time in the marine atmosphere zone, they are still scarce and there is no consensus concerning the best function to represent *Cs* behaviour. Moreover, the relationship between *Cs* and the availability of chlorides in the atmosphere has not been modelled. This work contributes to this discussion and analyses the behaviour of *Cs* in concrete exposed over 12.5 years in a marine atmosphere zone located in the northeast of Brazil, and analytical models to represent the relationships between *Cs*, time and chloride presence in the atmosphere are proposed. This is part of a long-term project studying chloride ion transportation into concrete under field exposure in the marine atmosphere zone.

## 2. Experimental Work

The experimental work carried out in this study took place in two steps: environmental characterisation and chloride concentration measurements from the concrete surface.

### 2.1. Environmental Characterization

The environmental characterization was undertaken with temperature, relative humidity, rainfall, wind characteristics and sea-salt data. Climatic data were collected by a Brazilian government weather station located in the region where the research took place. Sea-salt data were collected at location 10, 100, 200 and 500 m away from the sea (Figure 2) using the wet-candle method and a standardized capturing device, following the ASTM standard G140 [15]. With regard to long-term exposure, chloride deposition measurements were taken monthly in the first 2 years and, afterwards, in two additional cycles of 12 month monitoring following 5 and 10 years of the first monitoring cycle. No significant differences could be observed among these cycles. As a consequence, these results are represented by an average monthly rate of chloride deposition and its standard deviation.

### 2.2. Surface Chloride Concentration in Concrete

Prismatic concrete specimens ($0.15 \times 0.15 \times 1.40$ m$^3$) were cast using filler-modified Portland Brazilian cement, the chemical and physical properties of which are presented in Table 2. The main oxides were determined using XRF analysis and the insoluble residue, loss on ignition, specific surface and specific density were obtained using the Brazilian standards specific for each of these properties. The coarse aggregate was granitic crushed rock with a maximum diameter of 19 mm and the fine aggregate was river sand with a maximum diameter of 4.8 mm. Their granulometric curves are presented in Figure 3.

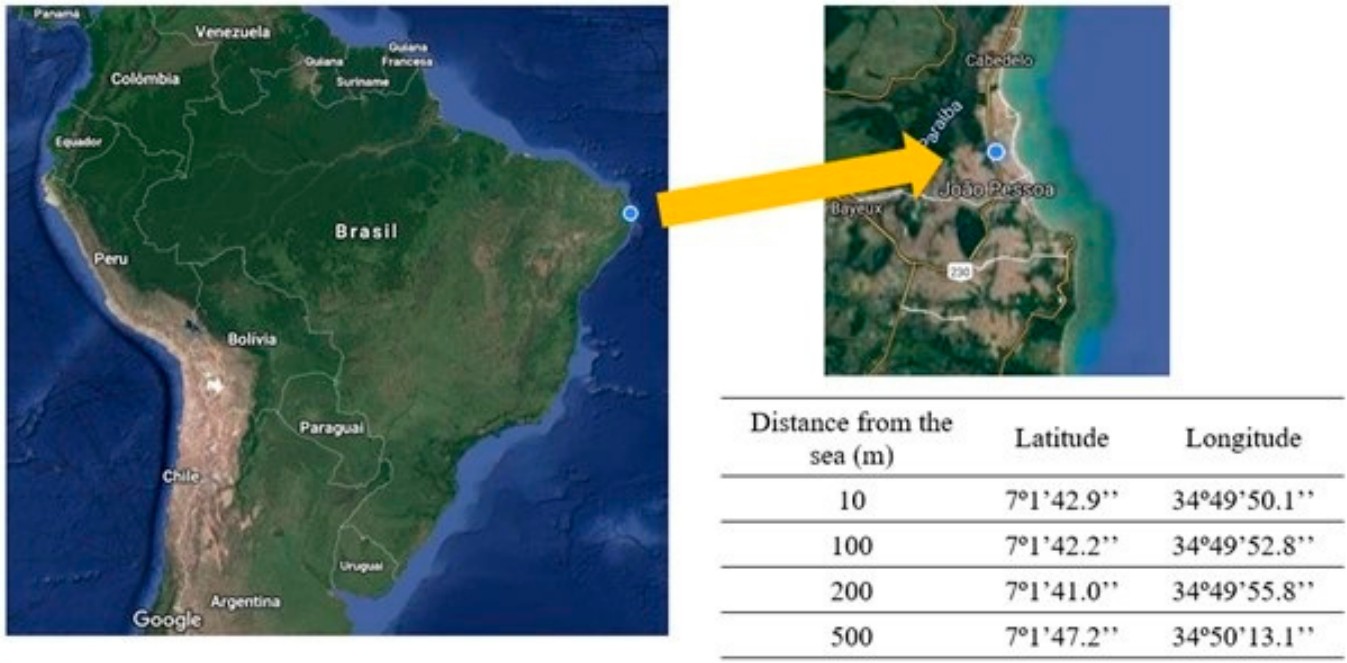

**Figure 2.** Region where wet-candle devices and concrete specimens were exposed.

**Table 2.** Chemical and physical properties of studied cement.

| Composition (%) | SO$_3$ | SiO$_2$ | Al$_2$O$_3$ | Fe$_2$O$_3$ | CaO | MgO | Na$_2$O | K$_2$O | Insoluble Residue (IR) | Loss on Ignition (LI) |
|---|---|---|---|---|---|---|---|---|---|---|
| | 3.21 | 18.11 | 4.31 | 2.27 | 59.87 | 3.61 | 0.21 | 1.51 | 1.45 | 5.50 |
| Property | Specific surface (cm$^2$/g) | | Specific density (g/cm$^3$) | | | | | | | |
| | 3650 | | 3.06 | | | | | | | |

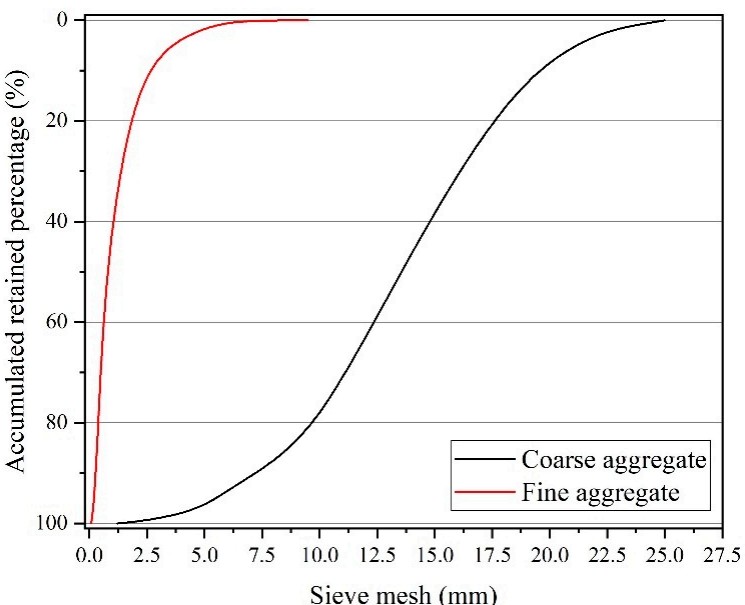

**Figure 3.** Granulometric curves for coarse and fine aggregates.

The concrete mixtures, which had w/b ratios of 0.65, 0.57 and 0.50, along with their physical properties are presented in Table 3. They are denoted C65, C57 and C50. As

these mixtures were produced several years ago, they have w/b ratios that are not typical nowadays but which represent a significant number of the reinforced concrete structures built in the past. Two samples for each test were used to characterise the concrete, as well as the studied cement and aggregates.

**Table 3.** Concrete mixtures and properties.

| Concrete | C50 | C57 | C65 |
|---|---|---|---|
| Mixture | | | |
| Cement (kg/m$^3$) | 406 | 356 | 320 |
| Sand (kg/m$^3$) | 769 | 812 | 840 |
| Coarse aggregate (kg/m$^3$) | 947 | 947 | 947 |
| Plasticiser (kg/m$^3$) | 1.22 | 1.06 | - |
| w/b | 0.5 | 0.57 | 0.65 |
| Property | | | |
| Slump (mm) | 80 | 80 | 80 |
| Compressive strength (MPa—28 days) | 31 | 27 | 20 |
| Concrete porosity (% volume—90 days) | 11.0 | 12.4 | 13.2 |

The specimens were cured in a wet chamber (RH > 95 %) for 7 days. The curing period of 7 days was adopted to represent a condition more similar to the building site. Afterwards, the specimens were painted with a waterproof film at those surfaces thorough which chloride penetration should be avoided; i.e., all the surfaces except the one facing the sea. Then, the specimens were placed at the same monitoring stations used for the wet-candle devices in an unsheltered condition.

After 6, 10, 14, 18, 46, 78 and 150 months of exposure, powdered samples were extracted from the specimens to obtain chloride profiles for the concrete. These samples were obtained by progressively grinding the concrete layers (from surface to bulk) of cores previously extracted from the prismatic concrete specimens. These cores had 75 mm diameters and 150 mm lengths. Although chloride profiles were obtained at each sampling period, only the surface chloride contents are analysed here. These samples were extracted from the first millimetre of the exposure surfaces of the specimens; i.e., a concrete layer from the surface to a depth of 1 mm. This option was adopted considering the two-zone profile that is characteristic of marine atmosphere zones, composed of a convection zone (outer zone) and a diffusion zone (inner zone) [16,17]. From this kind of profile, the extrapolation of the diffusion zone gives a theoretical *Cs* that is related to the internal peak that separates these two zones but not to the real concrete surface. Thus, directly obtaining a sample from the first concrete millimetre was considered to be closer to the real condition.

After obtaining the powdered samples, the total chloride content was determined through potentiometric titration using an automatic titrator from Metrohm and a double-junction electrode with a silver ring. The sample preparation followed the procedures of the International Union of Laboratories and Experts in Construction Materials, Systems and Structures [18].

## 3. Results and Discussion

### 3.1. Environmental Parameters

The climatic results showed that the temperature ranged between roughly 18 and 33 °C over this period, with an average value of 27 °C (Figure 4a). The relative humidity presented fluctuations between 62 and 99%, with an average value of 76.6% (Figure 4b). Higher values were reached during the winter (rainy season), which mainly takes place between May and August. The monthly average wind speed data ranged between 1.8 and 4.2 m/s, with an average value in the exposure period of around 3 m/s (Figure 5a). The predominant average wind directions remained between the east (E) and southeast (SE) directions (Figure 5b).

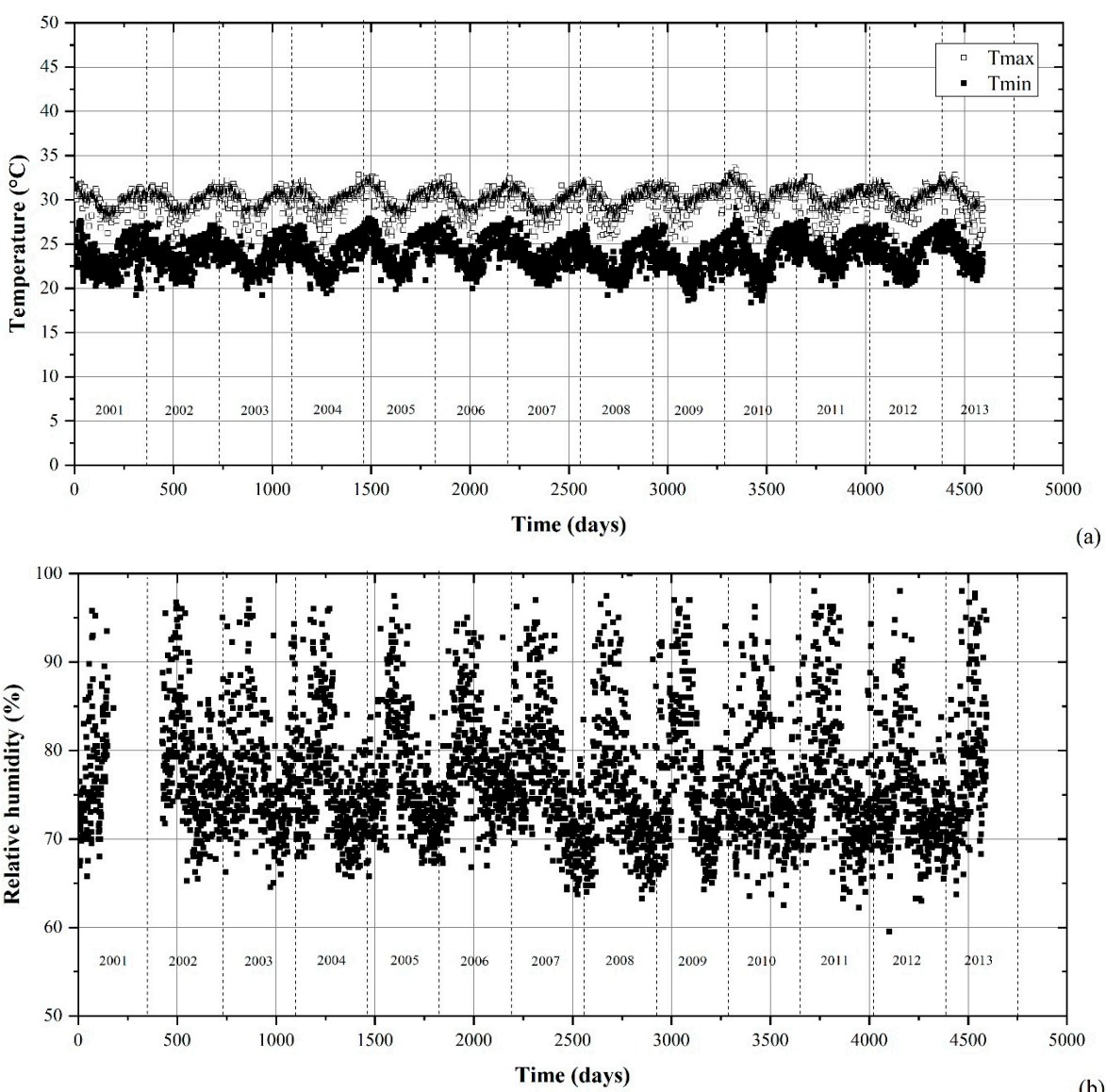

**Figure 4.** Maximum and minimum daily temperature (**a**) and daily average relative humidity (**b**) in the exposure region.

Average chloride deposition data are presented in Figure 6. This figure shows a strong decrease in salinity in the areas closest to the sea, which impacts chloride transportation into concrete at different levels. This drop in salinity is a consequence of the removal of marine aerosol salt particles, mainly due to the gravimetric effect. The gravimetric effect results in larger salt particles moving from the upper to the lower atmosphere layers until they are deposited on the ground, while marine aerosol is transported inland [19–21]. Other removal mechanisms can simultaneously act alongside the gravimetric effect and increase salt particle removal, such as the presence of obstacles and rainfall removal. However, the gravimetric effect assumes a greater significance than the other mechanisms [20].

The rate of the decrease in marine aerosol salinity when moving inland changed from site to site and was strongly dependent on wind characteristics [22]. However, a significant decrease in salinity when moving inland was observed in the vast majority of cases. As a result, concrete structures placed at different distances from the sea were subjected to different levels of aggressiveness.

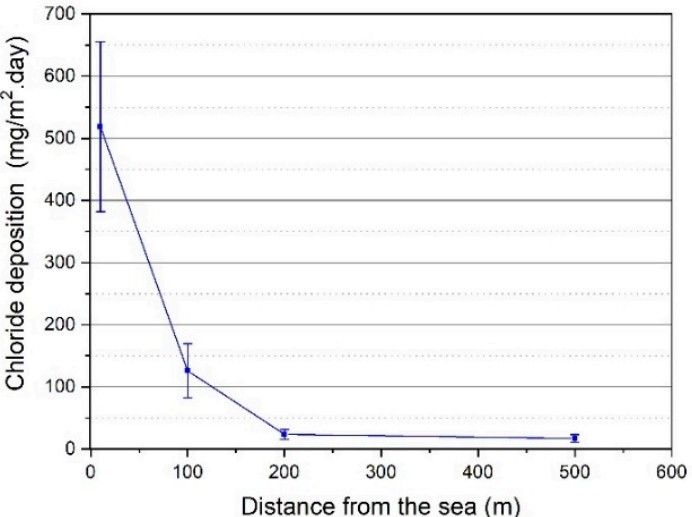

**Figure 5.** Monthly average wind speed (**a**) and monthly average wind direction (**b**).

**Figure 6.** Average chloride deposition data for the research period.

### 3.2. Surface Chloride Concentration and Its Relationship with Exposure Time

The results for the surface chloride concentration are presented in Figure 7, considering the three different concretes, the four exposure sites and their distances from the shoreline. Regarding the general aspects of the data, they show some fluctuation in the first months of exposure, followed by a period of significant increase and, afterwards, a tendency to increase at lower rates. These results suggest a tendency to reach a maximum over time, but it not possible to fully observe this in these 12.5 years of exposure. The initial fluctuation in the chloride concentration on the concrete surface can be attributed to environmental interaction, in which rainfall can play an important role, changing the chloride concentration on concrete surfaces with a relative higher magnitude [12,23].

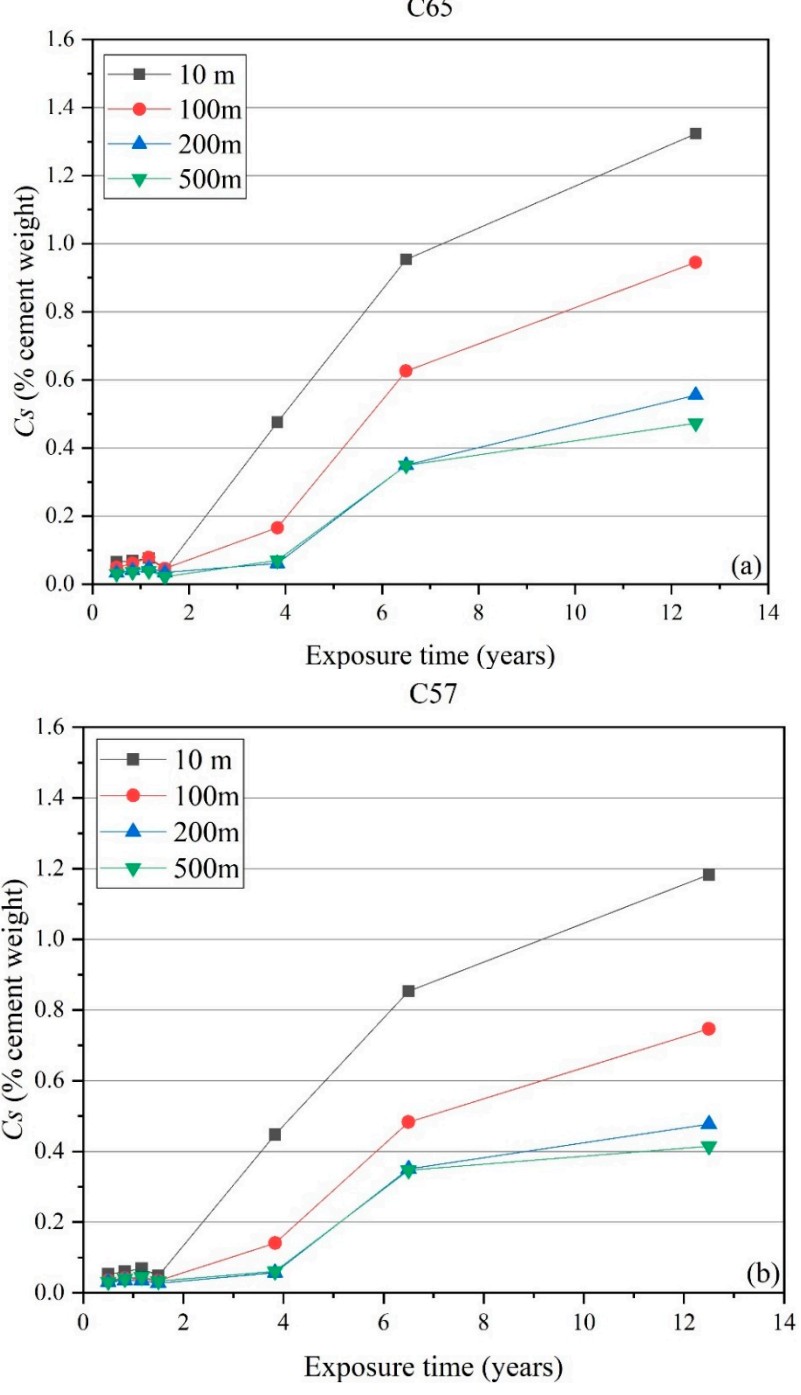

**Figure 7.** *Cont.*

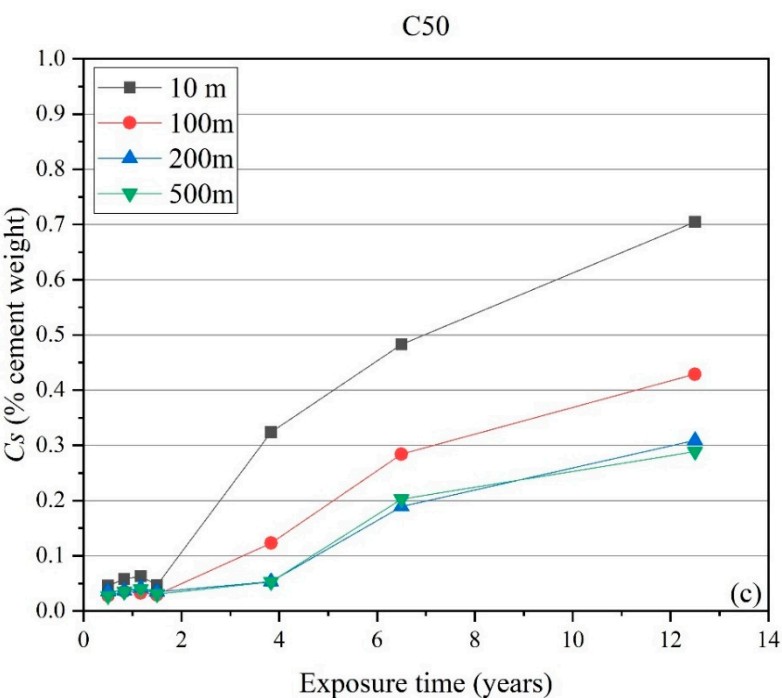

**Figure 7.** Relationship between *Cs* and exposure time for C65 (**a**), C57 (**b**) and C50 (**c**) concrete.

Regarding the influence of the concrete's characteristics, it is clear there was greater accumulation of chlorides on the concrete surface as concrete porosity increased (see Table 3), which was related to the greater availability of space for the accumulation of chloride ions. Furthermore, taking into account the distance from the sea, it is noticeable that the chloride concentration decayed as the distance from the sea increased, which was a consequence of the decrease in aggressiveness at sites far from the shoreline due to the lower availability of chlorides in the atmosphere, as can be observed in Figure 6. This decay did not follow a linear relationship, as can be indirectly observed in Figure 7 (and in the following figures), where it is possible to observe rates of decrease differing from one exposure site to another. This difference occurred because only some of the chlorides in atmosphere were deposited and captured on the concrete surface.

Exponential or power functions are usually used to represent the increase in *Cs* over time, as presented in Table 1. Nevertheless, the function that best fit the present experimental data was a sigmoidal function that considered the initial stage of fluctuation at low chloride concentrations, the sharpest growth period and the final period of stabilization. This can be better seen when comparing the determination coefficients obtained when using power functions and sigmoidal functions (Figure 8). This latter function is an alternative to the options presented in the literature that better represents the three stages of *Cs* growth over time.

After producing simulations using the functions adopted in Figure 8, it was possible to compare *Cs* values after 50 years of exposure in the marine atmosphere zone (Table 4). These results show that the sigmoidal function tends toward stabilization over a shorter time than the power function (as can also be seen in Figure 8) and that the values of *Cs* at the end of the 50 years may be between 3.7 and 7 times higher for the power functions. This overestimation was more pronounced for lower chloride concentrations in the atmosphere and less porous concretes. Considering the arrangement of the data in Figure 8 and previously published papers focusing on marine exposures [5], it is more reliable to expect a stabilization tendency after 10 years, which is better represented by the sigmoidal function.

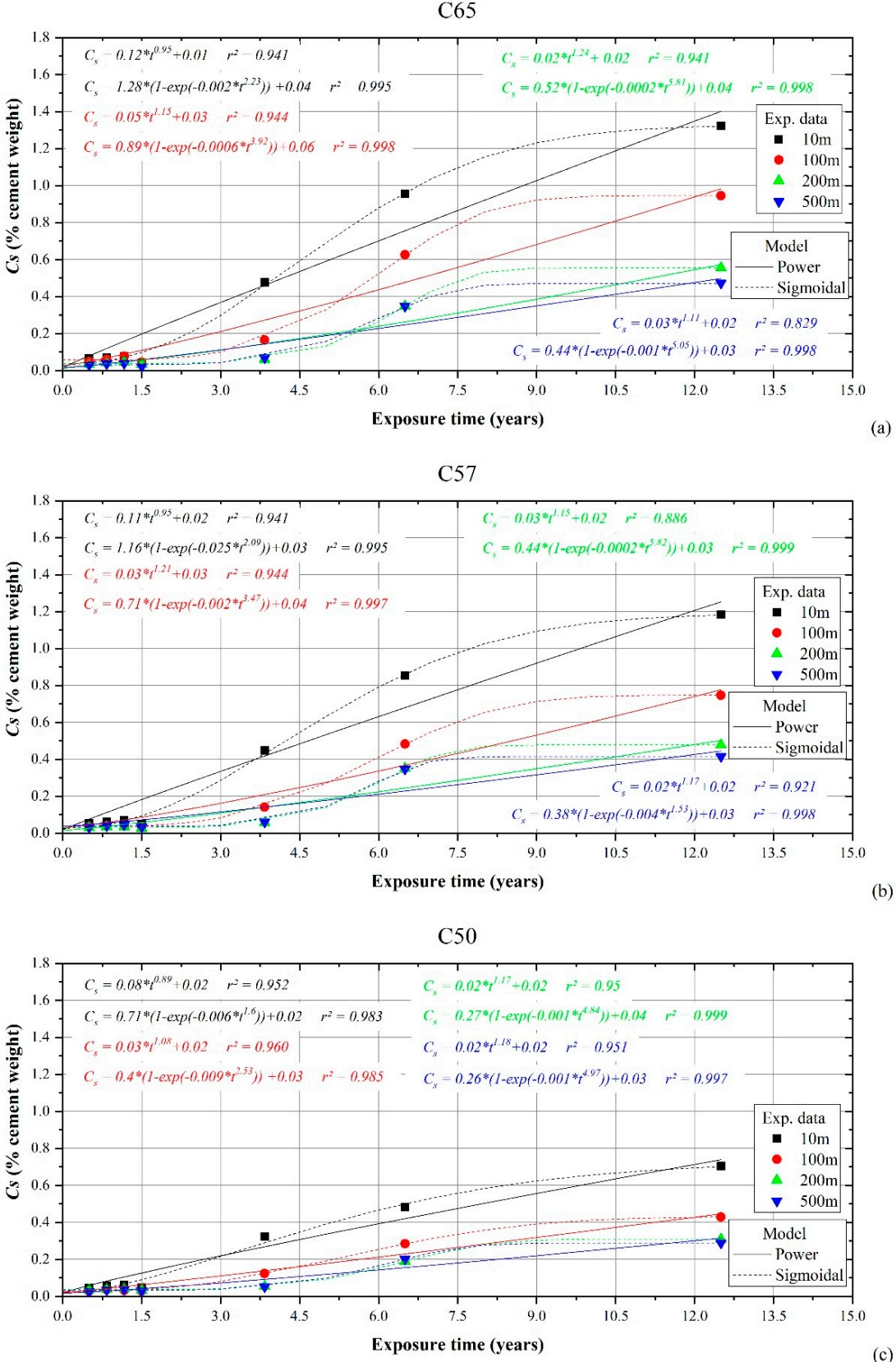

**Figure 8.** Fitting results with power and sigmoidal functions and *Cs* data for C65 (**a**), C57 (**b**) and C50 (**c**) concrete.

**Table 4.** Simulation of *Cs* values using power and sigmoidal functions for 50 years of exposure.

| Function | Concrete | Distance from the Sea | | | |
|---|---|---|---|---|---|
| | | **10 m** | **100 m** | **200 m** | **500 m** |
| | | *Cs* (% Cement Weight)—After 50 Years | | | |
| Power function | C65 | 4.94 | 4.52 | 2.58 | 2.33 |
| | C57 | 4.54 | 3.44 | 2.72 | 1.97 |
| | C50 | 2.62 | 2.07 | 1.96 | 2.04 |
| Sigmodal function | C65 | 1.32 | 0.95 | 0.56 | 0.47 |
| | C57 | 1.19 | 0.75 | 0.47 | 0.33 |
| | C50 | 0.70 | 0.43 | 0.31 | 0.29 |

Figure 9 gathers the literature data and the presented experimental data and shows the curves that represent each dataset. From this figure, it can be seen that the present results are from long-term field exposure and, thus, can catch part of the stabilization period. Furthermore, the literature data based on shorter exposure periods generally presented higher tendencies for increases in *Cs*. This can be explained not only by the fact that these studies were based on the first and second stages of the increase in *Cs* during the exposure time but also by the differences in the availability of chlorides in atmosphere. Nevertheless, the present data and Ghods et al. [7] data show good harmony, which may indicate similarities in environmental characteristics.

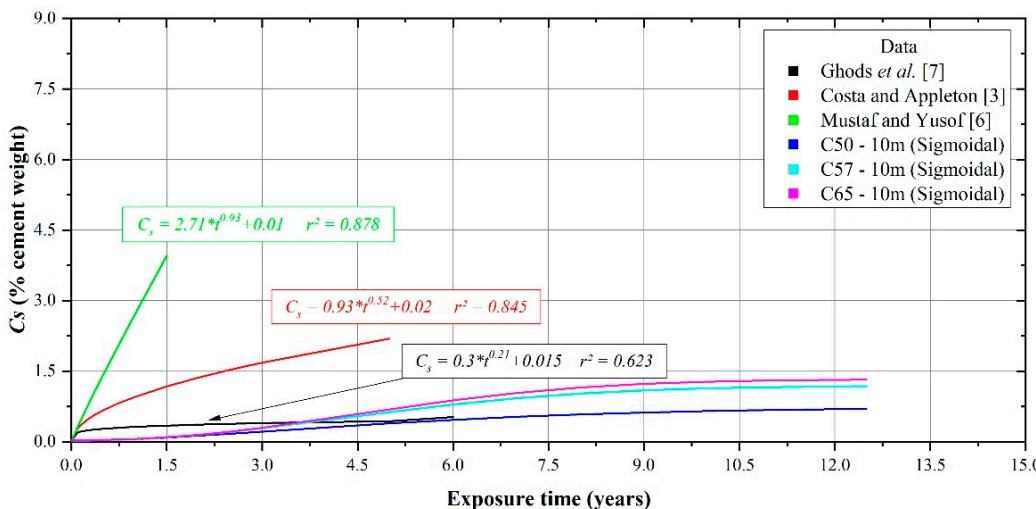

**Figure 9.** Fitting curves representing the literature data and the data from the present work [3,6,7].

### 3.3. Surface Chloride Concentration and Its Relationship with Chloride Deposition on a Wet Candle

Considering that the exposure time is not the main variable that influences the increase in *Cs* but rather the availability of chlorides in the atmosphere, the relationship between *Cs* and the accumulated deposition of chlorides on a wet candle (*Dac*) was analysed, which was obtained by summing the chloride deposition on the wet candle month-to-month (Figure 10).The power and sigmoidal functions were fitted to the experimental data in a similar way.

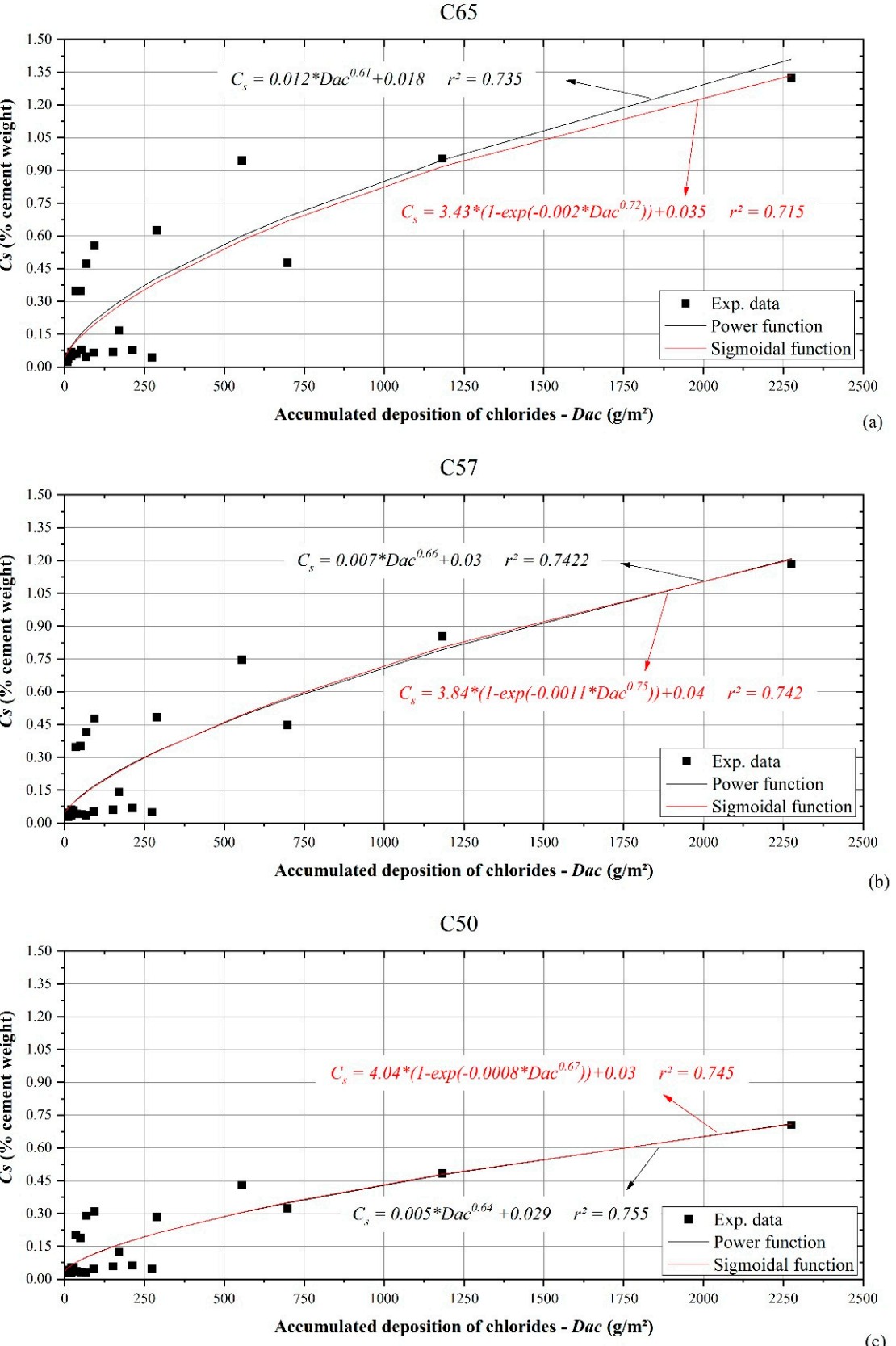

**Figure 10.** Relationship between *Cs* and accumulated deposition of chlorides (*Dac*) for C65 (**a**), C57 (**b**) and C50 (**c**) concrete.

As can be seen in Figure 10, there were no significant differences between the fitting results using the power and sigmoidal functions. This can be explained by the fact that, when using *Dac* values, data from different atmosphere salinities are considered together and, thus, there is an increase in the data dispersion in the region of low *Dac* values, making it more difficult to identify the first two stages of *Cs* growth. As a result, considering the simplicity of the model, the relationship between *Cs* and *Dac* can be represented by Equation (1), where *Cs* is the surface chloride concentration, $C_0$ is the initial chloride concentration in concrete, $k_{cs}$ is a coefficient associated with the concrete's ability to capture chlorides from atmosphere, *Dac* is the accumulated deposition of chlorides and *n* is a coefficient associated with the rate of the increase in *Dac* over time:

$$Cs = C_0 + k_{cs}(Dac)^n \tag{1}$$

This equation has the advantage of taking into account the direct relation between the availability of chlorides in the atmosphere and those captured on the concrete surface. Thus, it seems to be more suitable than those considered in previous studies [3,8,11].

## 4. Conclusions

In the present study, experimental data were collected over 150 months; i.e., a long-term field exposure period that could help in understanding the behaviour of *Cs* under long-term exposure. Therefore, the conclusions presented here may be considered relevant for the abovementioned exposure period and similar environments. The main conclusions are as follows:

(1) The increase in *Cs* with exposure time followed three stages: a first short stage characterised by an initial dispersion, followed by a period of increase and then a final period of stabilisation. Considering that the present data focused on the marine atmosphere zone, although the final period of stabilisation tended to reach a maximum, this condition was not fully reached in 12.5 years of exposure, indicating that more time is needed to reach the stabilisation condition in the marine atmosphere zone.

(2) Considering the fact that the concrete porosity at the surface layer may change depending on the w/b of concrete, and that *Cs* refers to the chloride concentration in the first millimetre of the concrete, the influence of concrete porosity on *Cs* behaviour occurred in a direct way. Concrete with higher w/b and, thus, higher porosity presented a stronger increase in *Cs*, independently of the exposure site.

(3) Chloride concentration in the atmosphere plays an important role in *Cs* behaviour. Higher availability of chlorides means higher *Cs* values. This leads to different curves for the increase in *Cs* depending on the availability of chlorides in the atmosphere at different distances from the sea. In terms of the exposure time, this behaviour can be represented by a power function or a sigmoidal function, with a better fit for the latter.

(4) Regarding the relationship between *Cs* and the availability of chlorides in the atmosphere, the function that best represented this relationship was $Cs = C_0 + k_{cs}(Dac)^n$, where *Cs* is the surface chloride concentration, $C_0$ is the initial chloride concentration in concrete, $k_{cs}$ is a coefficient associated with the concrete's ability to capture chlorides from the atmosphere, *Dac* is the accumulated deposition of chlorides and *n* is a coefficient associated with the rate of the increase in *Dac* over time.

The continuation of this study will make it possible to obtain data from even longer exposure periods, which may contribute to the refinement of the models presented here and the evaluation of other modelling approaches, such as numerical ones.

**Author Contributions:** G.R.M.: Conceptualization, methodology, funding acquisition, supervision, experimental work, formal analysis and writing. P.R.F.: Experimental work, formal analysis and writing. C.A.: Conceptualization and methodology. All authors have read and agreed to the published version of the manuscript.

**Funding:** This research was partially funded by the Brazilian National Council for Scientific and Technological Development (CNPq), grant number 305706/20160.

**Acknowledgments:** The authors thank the Brazilian National Council for Scientific and Technological Development (CNPq) and the Research Laboratory on Building and Waste Materials of IFPB (Paraíba Federal Institute) for partially supporting this research. The authors also thank the Brazilian Institute of Meteorology for providing climatic data.

**Conflicts of Interest:** The authors declare no conflict of interest.

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
