# Peer review of "Long-Term Chloride Accumulation on Concrete Surface in Marine Atmosphere Zone—Modelling the Influence of Exposure Time and Chloride Availability in Atmosphere"

_cmd, doi:10.3390/cmd3030021_

Round 1

Reviewer 1 Report

The manuscript entitled "Long-term chloride accumulation on concrete surface in marine atmosphere zone – modelling the influence of exposure time and chloride availability in atmosphere " presents an interesting experimental study conducted on the chloride accumulation on concrete surface in the areas close to the ocean shore. However, the number of tested samples isn’t presented and a few other issues must be addressed. The paper needs minor revisions before it is processed further, some comments follow:

·       The title seems to be too long. Please consider replacing the title with a clear formula that reflects the content of the manuscript.

·       The introduction section could be improved. The reviewed literature is very OLD, only 2 studies published after 2010 (1 from 2016 and 1 from 2017) have been considered (this is way too low). Please conduct a literature survey of the most recent published studies and consider those the newest discoveries in the field. Please improve the introduction section and the discussions considering these studies.

·       Table 2 - two types of iron oxides have been detected in this type of material, therefore, please replace Fe2O3 with FexOy or provide the scientific proof to support your results. Moreover, which methods have been used to evaluate the properties presented in the table? Are these data obtained by the authors or they have been provided by the manufacturer (please introduce corresponding comments into the manuscript).

·       Table 3 – please provide particle size distribution or standard evaluation of sand and coarse aggregates. To assure the experiments' repeatability it is important to clearly and exhaustively describe each material and parameter used in the experiments.

·       The clarity of the microstructures from figure 1 is unclear, please provide higher resolution images and increase the size of those two captions to allow to the reader to make a clear correlation between the affirmations from the figure and its description.

·       Tables 2 and 3. How many samples have been tested? What is the standard deviation value for each measurement?

·       Please provide some future directions and limitations of the study.

·       Figure 3 and Figure 4 – please provide a trend line.

·       How many samples have been tested from each batch to evaluate the Relationship between Cs and exposure time for concretes? What was the standard deviation value of each measurement? As can be seen from figure 6, the lines for 200m and 500m are very similar. However, if multiple measurements are considered and the mean value will be plotted, the trend lines will be the same?

·       Please improve the conclusions and present them following the main recommendations by Academia of giving the conclusions of the study by points with highlights.

Author Response

Comments (responses in red):

The title seems to be too long. Please consider replacing the title with a clear formula that reflects the content of the manuscript.

Thanks for your comment. The title is really too long. However, it gives the information about the whole contribution of the paper. This way, we think that maintaining the title as it is does not represent a devaluation for the paper

The introduction section could be improved. The reviewed literature is very OLD, only 2 studies published after 2010 (1 from 2016 and 1 from 2017) have been considered (this is way too low). Please conduct a literature survey of the most recent published studies and consider those the newest discoveries in the field. Please improve the introduction section and the discussions considering these studies.

Thanks for your comment. The paper is focused on the evolution of Cs along time specifically in the marine atmosphere zone. As commented in Introduction section, this topic was less studied, mainly when considering the marine atmosphere zone. There are newest papers that consider other marine zones (submerged, tidal or splash zones), but not the marine atmosphere zone and for those zones the evolution of Cs along time might be different. This is the reason for not incorporating other papers in the literature review.

Table 2 - two types of iron oxides have been detected in this type of material, therefore, please replace Fe2O3 with FexOy or provide the scientific proof to support your results. Moreover, which methods have been used to evaluate the properties presented in the table? Are these data obtained by the authors or they have been provided by the manufacturer (please introduce corresponding comments into the manuscript).

Thanks for your comment. Additional comments on how test results were obtained were incorporated to the manuscript.

Table 3 – please provide particle size distribution or standard evaluation of sand and coarse aggregates. To assure the experiments' repeatability it is important to clearly and exhaustively describe each material and parameter used in the experiments.

Thanks for your comment. Aggregates characteristics were added to the manuscript (text and Figure 3)

The clarity of the microstructures from figure 1 is unclear, please provide higher resolution images and increase the size of those two captions to allow to the reader to make a clear correlation between the affirmations from the figure and its description.

Thanks for your comment. I am not sure if the reviewer was really talking about the Figure 1. Anyway, it was enlarged as much as possible to make its information easier to see.

Tables 2 and 3. How many samples have been tested? What is the standard deviation value for each measurement?

Thanks for your comment. The number of samples was informed, as well as that the presented data represent average results.

Please provide some future directions and limitations of the study.

Thanks for your comment. A paragraph about the limitations and directions of this study was added in the Conclusions section.

Figure 3 and Figure 4 – please provide a trend line.

Thanks for you comment. Figures 3 and 4 represent climatic parameters from the exposure region. They were introduced in the paper to qualify the field exposure region. Including trend lines for these parameters do not represent the aim of the paper and, in some cases, it is not feasible to do it due to the number of influencing variables. We respect the suggestion from the reviewer but, in this case, we decided to maintain the Figures without changes.

How many samples have been tested from each batch to evaluate the relationship between Cs and exposure time for concretes? What was the standard deviation value of each measurement? As can be seen from figure 6, the lines for 200m and 500m are very similar. However, if multiple measurements are considered and the mean value will be plotted, the trend lines will be the same?

Thanks for your comment. Samples from field exposure were obtained from individual cores extracted from piles. They represent an exposure surface of about 45 cm2.

Please improve the conclusions and present them following the main recommendations by Academia of giving the conclusions of the study by points with highlights.

Thanks for your comment. This comment was addressed, and the conclusions section was improved

Reviewer 2 Report

The manuscript entitled "Long-term chloride accumulation on concrete surface in marine atmosphere zone–modelling the influence of exposure time and chloride availability in atmosphere carried out experimental work to investigate the environmental characterization and chloride concentration measurements on the concrete surface considering three different concrete mixtures.

Comments:

1-     Technical writing is not good to be accepted. Therefore, it should be improved throughout the manuscript.

2-     The abstract did not highlight the problem and a little about the used methodology.

3-     Line 89: "based in" should be "based on".

4-     Section 2: More information and illustration should be provided on how to obtain chloride profiles in concrete.

5-     Figure 6: The format of numbers on the vertical axes in this Figure should be corrected. The decimal points are not correct. This should be corrected throughout the manuscript.

6-     Figure 7: Is it possible to consider the w/b or type of concrete as a variable in the proposed equations?

7-     Line 203: I think the expression "de data" is not English.

8-    Quantitative analysis should be provided in the discussion and conclusion.

Author Response

Comments (responses are in red):

Technical writing is not good to be accepted. Therefore, it should be improved throughout the manuscript.

Thanks for your comment. Another revision was carried out.

The abstract did not highlight the problem and a little about the used methodology.

Thanks for your comment. The problem was highlighted and more information about the methodology was added to the abstract.

Line 89: "based in" should be "based on".

Thanks for your comment. The text was corrected.

Section 2: More information and illustration should be provided on how to obtain chloride profiles in concrete.

Thanks for your comment. More information about the obtaining chloride profiles was added in section 2.

Figure 6: The format of numbers on the vertical axes in this Figure should be corrected. The decimal points are not correct. This should be corrected throughout the manuscript.

Thanks for your comment. Numbers in Figure 6 and thorough the text were corrected.

Figure 7: Is it possible to consider the w/b or type of concrete as a variable in the proposed equations?

Thanks for your comment. The type of concrete was considered when trend lines were obtained for each concrete in this Figure. A general model considering this other variable was not possible to be considered at this moment.

Line 203: I think the expression "de data" is not English.

Thanks for your comment. Figure 8 caption was corrected.

Quantitative analysis should be provided in the discussion and conclusion.

Thanks for your comment. A quantitative analysis considering concrete characteristics influence on Cs was added after Figure 7, which was also considered in conclusion section.

Reviewer 3 Report

This work should be of interest to those working in the field. It can be published provided authors can satisfactorily correct, improve or justify the following points.

Why only 7 days of curing?

Line 112. The samples “were painted with a waterproof film at those surfaces thorough which chloride penetration should be avoided.”. Which surfaces? One, more?

How were the samples placed in the testing sites? In sheltered or unsheltered conditions (i.e., exposed to rain and its washing conditions or not)?  Disposition of the uncoated surfaces (placed facing up or normal to the floor)? At which distance from the ground?

Line 119. “These samples were extracted in the first millimetre of the exposure surfaces of specimens.” More information about the sampling is necessary. Explain how it was done in more detail.  Was there a layer of salt on the surface or most of the salt was inside the specimens?

This sampling procedure determines to great extent the results of Cs, particularly because the chloride is given with respect to the cement. This seems to be the weakest part of the work because it might not be comparable to other works. A smaller or larger amount/thickness of the sample will affect the results, the reproducibility, and the comparison among the different samples and with the results from others.

Thinking of reproducibility, how many replicate samples were tested in each site and for each time?

Line 120. Give details about the chloride potentiometric titration: equipment, selective electrode and a small account of the pertinent conditions of RILEM. Include the number of the reference.

Regarding Fig. 5 it is important (mandatory) to show the chloride deposition in a month basis in the 4 distances from the sea during the 12.5 years of testing (in a similar way as the other atmospheric parameters). This is a key parameter in this paper and such a plot is of extremely importance.

In Figure 6 the values correspond to the amount of chloride to cement weight? The cement in the 1 mm thickness that was removed for testing? Amount of Chloride or NaCl?

Was a layer of salt on the surface or most of the salt was inside the specimens?

How much chloride on the surface (outside) and how much in the first 1 mm inside the sample? Porosity influences more the outer or the inner part?

Wouldn’t have been better to simply brush/wash the salt layer at the surface of the samples? What was the advantage to include 1 mm of sample?

Figure 6 (Y axis): Point not comma.

From 10 to 100 m from the shore the amount of Cl is 3-4 times smaller. However, the chloride present in the samples is not more than 1-2 times larger. Why?

Figure 7. Why potential appears in the legend inside the graphs? Should not be power?

Maybe better than present the equations in the plots, consider putting them in a table.

Figure 8. Not indicated but it seems that the curves from this work shown in the figure correspond to the power function. Why? Considering that the sigmoidal function provided a better fit, shouldn’t it be shown instead?

Figure 9.

- Could be interesting to identify with different colours the points corresponding to different distances to the sea.

- Given the number of points that should have been measured, more points would be expected.

- One of the points shows 2.25 Kg of Cl (not NaCl?). Just one point with this condition?

Conclusions

“The influence of concrete porosity on Cs behaviour happens in a direct way. Concretes with higher w/b and thus higher porosity present a stronger increase of Cs independently of the exposure site.”

 The Effect of porosity is not clearly explained. How the porosity which is a bulk property affects Cs which is a surface parameter?

Author Response

Comments (responses in red):

Why only 7 days of curing?

Thanks for your comment. Seven days curing was justified in the text (section 2.2)

Line 112. The samples “were painted with a waterproof film at those surfaces thorough which chloride penetration should be avoided.”. Which surfaces? One, more?

How were the samples placed in the testing sites? In sheltered or unsheltered conditions (i.e., exposed to rain and its washing conditions or not)? Disposition of the uncoated surfaces (placed facing up or normal to the floor)? At which distance from the ground?

Thanks for your comment. More details about painted faces of the specimens and about the field exposure condition were added to the text (section 2.2)

Line 119. “These samples were extracted in the first millimetre of the exposure surfaces of specimens.” More information about the sampling is necessary. Explain how it was done in more detail. Was there a layer of salt on the surface or most of the salt was inside the specimens?

This sampling procedure determines to great extent the results of Cs, particularly because the chloride is given with respect to the cement. This seems to be the weakest part of the work because it might not be comparable to other works. A smaller or larger amount/thickness of the sample will affect the results, the reproducibility, and the comparison among the different samples and with the results from others.

Thanks for your comment. In general, there are two ways adopted in literature to obtain Cs data: extrapolations from chloride profiles or directly obtaining the surface chloride concentration from an outer layer of concrete that usually is about 1 mm thick. The option of this work was the second one and this was better explained in the paper.

Thinking of reproducibility, how many replicate samples were tested in each site and for each time?

Line 120. Give details about the chloride potentiometric titration: equipment, selective electrode and a small account of the pertinent conditions of RILEM. Include the number of the reference.

Thanks for your comment. Samples from field exposure were obtained from individual cores extracted from piles. They represent an exposure surface of about 45 cm2. These details were provided in section 2.2, as well as more information about the potentimetric titration.

Regarding Fig. 5 it is important (mandatory) to show the chloride deposition in a month basis in the 4 distances from the sea during the 12.5 years of testing (in a similar way as the other atmospheric parameters). This is a key parameter in this paper and such a plot is of extremely importance.

Thanks for your comment. Chloride deposition on wet candle was not measured in all months along the 12.5 years. ASTM, ISO and Brazilian standards consider that an environment can be characterised with results continuously obtained from 12 months. In the present case, considering the long-term exposure, the first 2 years were monitored and afterwards two additional cycles of 12 month monitoring were carried out after about 5 and 10 years of the first monitoring cycle. No significant differences could be observed among these cycles. This information was added in section 2.2.

In Figure 6 the values correspond to the amount of chloride to cement weight? The cement in the 1 mm thickness that was removed for testing? Amount of Chloride or NaCl?

Was a layer of salt on the surface or most of the salt was inside the specimens?

How much chloride on the surface (outside) and how much in the first 1 mm inside the sample? Porosity influences more the outer or the inner part?

Wouldn’t have been better to simply brush/wash the salt layer at the surface of the samples? What was the advantage to include 1 mm of sample?

Thanks for your comment. In general, there are two ways adopted in literature to obtain Cs data: extrapolations from chloride profiles or directly obtaining the surface chloride concentration from an outer layer of concrete that usually is about 1 mm thick. The advantage of using 1 mm of sample is that when brushing the surface only free and not fixed chlorides could be detected, which is certainly far from the shape of the chloride profile in the convection zone. Considering the first millimetre, it is true that we are not representing the zero depth, but in our opinion it is much more realistic and fits better with the profile in the convection zone.

Figure 6 (Y axis): Point not comma.

Thanks for your comment. It was corrected.

From 10 to 100 m from the shore the amount of Cl is 3-4 times smaller. However, the chloride present in the samples is not more than 1-2 times larger. Why?

Thanks for your comment. At 100 m from the shoreline, chloride deposition is about 25% of that observed at 10 m from the shoreline. The relation between chlorides from atmosphere and chlorides on concrete surface is not linear. It depends on the rate of chloride deposition on concrete surface that is different from wet candle and also on the rate of capturing chlorides on concrete surface. Not all chloride ions that impact on concrete surface will remain on it and even less will be fixed on surface.

Figure 7. Why potential appears in the legend inside the graphs? Should not be power? Maybe better than present the equations in the plots, consider putting them in a table.

Thanks for your comment. The function type was corrected to power function.

Figure 8. Not indicated but it seems that the curves from this work shown in the figure correspond to the power function. Why? Considering that the sigmoidal function provided a better fit, shouldn’t it be shown instead?

Thanks for your comment. It was changed to sigmoidal function.

Figure 9.

- Could be interesting to identify with different colours the points corresponding to different distances to the sea.

- Given the number of points that should have been measured, more points would be expected.

- One of the points shows 2.25 Kg of Cl (not NaCl?). Just one point with this condition?

Thanks for your comment. In Figure 9, the aim is not to show the influence of the distance from the sea on Cs, but the influence of chloride deposition on wet candle on Cs. This is the reason for not separating the data from each distance from the sea.

Although it cannot be visible, all the points are depicted in the graphs. As most of then are in the region of low concentrations, several of them are overlaid.

The point 2.25 kg/m2 of Cl represents an exposure time of 150 months and the Cs for concretes at 10 m from the sea. This is the reason for this point being isolated.

Conclusions

The influence of concrete porosity on Cs behaviour happens in a direct way. Concretes with higher w/b and thus higher porosity present a stronger increase of Cs independently of the exposure site.”

The Effect of porosity is not clearly explained. How the porosity which is a bulk property affects Cs which is a surface parameter?

Thanks for your comment. Concrete porosity is related to w/b and also to the region of concrete (surface or bulk). Regarding that at surface its porosity may change depending on the w/b of concrete and thus on the amount of chlorides that are accumulated in the first millimetre of concrete, it is visible that concrete with higher w/b and thus higher porosity present a stronger increase of Cs independently of the exposure site. This was better explained in Conclusions section. This information was complemented in the text.

Round 2

Reviewer 2 Report

The authors have addressed all the reviewer's comments and the manuscript can be accepted for publication.